# Circulating MicroRNA: Incident Asthma Prediction and Vitamin D Effect Modification

**DOI:** 10.3390/jpm11040307

**Published:** 2021-04-16

**Authors:** Jiang Li, Anshul Tiwari, Hooman Mirzakhani, Alberta L. Wang, Alvin T. Kho, Michael J. McGeachie, Augusto A. Litonjua, Scott T. Weiss, Kelan G. Tantisira

**Affiliations:** 1Research Center, The Seventh Affiliated Hospital of Sun Yat-Sen University, Shenzhen 518107, China; rejia@channing.harvard.edu; 2Channing Division of Network Medicine, Brigham and Women’s Hospital, Harvard Medical School, Boston, MA 02115, USA; reati@channing.harvard.edu (A.T.); rehom@channing.harvard.edu (H.M.); reawa@channing.harvard.edu (A.L.W.); alvin_kho@hms.harvard.edu (A.T.K.); remmg@channing.harvard.edu (M.J.M.); restw@channing.harvard.edu (S.T.W.); 3Computational Health Informatics Program, Boston Children’s Hospital, Boston, MA 02115, USA; 4Division of Pediatric Pulmonary Medicine, Golisano Children’s Hospital at Strong, University of Rochester Medical Center, Rochester, NY 14642, USA; augusto_litonjua@urmc.rochester.edu; 5Division of Pediatric Respiratory Medicine, Rady Children’s Hospital San Diego, University of California, San Diego, CA 92123, USA

**Keywords:** vitamin D, asthma, miRNA, circulating microRNA, biomarker

## Abstract

Of children with recurrent wheezing in early childhood, approximately half go on to develop asthma. MicroRNAs have been described as excellent non-invasive biomarkers due to their prognostic utility. We hypothesized that circulating microRNAs can predict incident asthma and that that prediction might be modified by vitamin D. We selected 75 participants with recurrent wheezing at 3 years old from the Vitamin D Antenatal Asthma Reduction Trial (VDAART). Plasma samples were collected at age 3 and sequenced for small RNA-Seq. The read counts were normalized and filtered by depth and coverage. Logistic regression was employed to associate miRNAs at age 3 with asthma status at age 5. While the overall effect of miRNA on asthma occurrence was weak, we identified 38 miRNAs with a significant interaction effect with vitamin D and 32 miRNAs with a significant main effect in the high vitamin D treatment group in VDAART. We validated the VDAART results in Project Viva for both the main effect and interaction effect. Meta-analysis was performed on both cohorts to obtain the combined effect and a logistic regression model was used to predict incident asthma at age 7 in Project Viva. Of the 23 overlapped miRNAs in the stratified and interaction analysis above, 9 miRNAs were replicated in Project Viva with strong effect size and remained in the meta-analysis of the two populations. The target genes of the 9 miRNAs were enriched for asthma-related Kyoto Encyclopedia of Genes and Genomes (KEGG) signaling pathways. Using logistic regression, microRNA hsa-miR-574-5p had a good prognostic ability for incident asthma prognosis with an area under the receiver operating characteristic (AUROC) of 0.83. In conclusion, miRNAs appear to be good biomarkers of incident asthma, but only when vitamin D level is considered.

## 1. Introduction

Asthma is a chronic inflammatory disease characterized by airway hyper-responsiveness, mucus hypersecretion, and airway remodeling [1]. According to the Centers for Disease Control and Prevention (CDC) report in 2018, about 25 million Americans have asthma, including 7.5% of children (age < 18 years) and 7.4% of adults (age 18+ y) [2]. Asthma is more common in children than adults and the number of cases with childhood asthma increased from 2014 to 2018 by 744,397 in the US [2]. The cost of asthma in direct and indirect medical care is estimated to exceed 18 billion dollars each year [3].

Recurrent wheeze in early childhood is a significant risk factor for asthma, but not all children who wheeze go on to develop asthma. According to epidemiologic studies, 60% of young wheezers are symptom-free at the age of 6 y, and a majority of them remain asymptomatic at the age of 11 and 16 [4,5,6]. Although wheezing in early childhood is common, finding a way to predict who will continue to wheeze and develop asthma is important for asthma diagnosis and management.

Vitamin D is a fat-soluble secosteroid that can be produced in response to the skin being exposed to sunlight but is present in the diet. Vitamin D is essential for bone growth and remodeling and is also involved in immune system function [7,8]. Currently, epidemiologic and experimental evidence has suggested an association between higher vitamin D levels in pregnant women and reduced asthma in their offspring, but the results of these observational studies differ in the results [9]. To address this problem, the Vitamin D Antenatal Asthma Reduction Trial (VDAART) was developed, as a randomized, double-blind, placebo-controlled trial with the primary outcome to determine whether prenatal vitamin D supplementation in pregnant women can decrease asthma incidence in the women’s offspring [10]. 

MiRNAs are small non-coding RNAs with a length of 21–24 bp and they can be loaded into RISC (RNA-induced Silencing Complex) to bind to the 5′-UTR of mRNAs and thus down-regulate gene expression [11]. MiRNAs also exist in a variety of biofluids (extracellular RNAs), such as plasma, serum, sputum, and urine [12]. With the development of next-generation sequencing, an increasing number of studies have profiled small RNA-Seq for miRNA expression [13,14] and miRNAs have been utilized as the non-invasive biomarkers of both diseases and treatment responses [13,15,16,17,18]. Some miRNAs have been identified to associate with childhood asthma in several studies [19,20,21,22]. 

In this study, we sought to identify circulating miRNAs that can predict incident childhood asthma and assess that prediction in relation to vitamin D treatment. 

## 2. Materials and Methods

### 2.1. Participant Selection

We selected 75 participants from the VDAART (ClinicalTrials.gov Identifier: NCT00920621) based on the presence of recurrent wheezing occurring at 3 years of age. These participants provided plasma samples at age 3 and were followed up at 5 years of age when their asthma status was diagnosed by physicians. Two participants were not followed up at age 5 years and were removed in the following analysis.

### 2.2. Small RNA Sequencing

Total RNA was isolated from samples by the Qiagen miRNAeasy Serum/Plasma extraction kit and QIAcube automation. Small RNA sequencing libraries were prepared using the Norgen Biotek Small RNA Library Prep Kit and then sequenced on the Illumina NextSeq 500 platform at 51 bp single-end reads. 

We used the exRNA-processing toolkit (exceRpt) to assess the read quality and profile the miRNAs [23]. Raw read counts were log-transformed and quantile normalized. Read counts less than 5 and miRNAs with coverage less than 80% of all samples were removed. 

### 2.3. Statistical Analysis

Statistical analysis was performed using R software version 3.6.3 (Bell Laboratories, Murray Hill, NJ, USA). Logistic regression was used to assess the association between the miRNAs normalized count and asthma status (asthma: Y = 1 and no asthma: Y = 0) at age 5 years for participants in VDAART. We first examined the association in all candidate participants and then in the interaction and stratified analyses by the vitamin D treatment status (high vitamin D treatment group (4400 international units (IU)/day): TG = 1 and low vitamin D treatment group (400 IU/day): TG = 0). 

### 2.4. Validation

We selected 20 participants in Project Viva (ClinicalTrials.gov Identifier: NCT02820402) a pre-birth cohort established to examine the association between prenatal diet and other factors with maternal and child health [24]. All of these 20 participants had a history of recurrent wheeze at 3 years old when the blood draw was taken. These plasma samples were prepared and sequenced in the same way as the VDAART samples, as well as having the miRNA profiling performed. 

The vitamin D concentration was measured by both an automated chemiluminescence immunoassay and a manual radioimmunoassay at 16–26 weeks’ gestation from each mother. We used the average of the two values to estimate the maternal 25(OH)D concentration [25] and divided mother-child pairs into adequate (maternal levels ≥ 20 ng/mL) and inadequate (maternal levels < 20 ng/mL) groups according to the report from the Institute of Medicine (IOM) [26]. For the 20 mother-child pairs in Project Viva, 12 children were from the adequate group and the other 8 children were from the inadequate group of maternal 25(OH)D concentration. The primary outcome was the asthma status at 7 years of age. We validated our results in the maternal 25(OH)D adequate group as well as considering the interaction by maternal vitamin D status. 

### 2.5. Meta-Analysis

To combine the results from both VDAART and Project Viva, we performed a meta-analysis of the two data sets. The R package “metafor” was employed to combine the odds ratio (OR) values via the restricted maximum-likelihood estimator [27]. 

### 2.6. MiRNA-Target Gene Network and Kyoto Encyclopedia of Genes and Genomes (KEGG) Pathway Enrichment Analysis

We built the miRNA-target genes interactions network and performed Kyoto Encyclopedia of Genes and Genomes (KEGG) pathway enrichment analysis through miRNet 2.0 (McGill University, Montreal, Quebec, Canada) [28]. MiRNA-target interactions (MTIs) came from miRTarBase release 8.0 [29]. Nominal *p* values of pathways were adjusted for false discovery rate (FDR). 

### 2.7. Prediction

We built a logistic regression model to predict asthma status at age 7 years in the Project Viva adequate vitamin D group. The area under the receiver operating characteristic (AUROC) was calculated to measure the performance of our logistic regression model in the prediction of asthma status.

## 3. Results

### 3.1. Baseline Characteristics

A total of 75 participants in VDAART were selected, of whom 31 were children with asthma at age 5 years, 42 were healthy controls and 2 were lost follow-up. The baseline characteristics of the 73 subjects with available outcome for asthma status at age 5 years were shown in Table 1. Of the 73 participants, 33 were in the treatment arm of the VDAART trial where the children were born to mothers who had received 4400 IU of vitamin D3 per day during pregnancy and 40 were in the placebo arm (control group) where the children were born to mothers who had received regular multivitamins containing 400 IU of vitamin D3 per day. Sex, race, and other characteristics were evaluated and found not to be different between treatment arms.

### 3.2. Main Analysis

We investigated the association between miRNAs at age 3 years and asthma status at age 5 years in all 73 VDAART participants and then examined the results in Project Viva. The results are shown in Appendix A. Six miRNAs were nominally significant (*p* < 0.05) in VDAART and three were present in Project Viva. Only miRNA hsa-miR-548k could be validated as being in the same direction of effect in VDAART and Project Viva. Given the marginal main effects, we performed interaction and stratified analyses on our dataset by intervention arm.

### 3.3. Significant miRNAs in Treatment Interactions Analysis

We identified 38 miRNAs that were associated with asthma at age 5 years in the interaction analysis (nominal *p* < 0.05) in VDAART. The results were ranked by effect size (OR) and shown in Table 2. The effect modification by vitamin D status was seen in both the positive and negative directions, with strong effect estimates noted. Sixteen miRNAs had OR values greater than 3 and five miRNAs had OR less than 0.33. Hsa-miR-3942-5p had the largest OR value of 178.73 [95% confidence interval (CI), 4.6 to 6947.95], a high risk of incident asthma and hsa-miR151a-5p had the smallest OR value of 0.05 [95% CI, 0.01 to 0.42], a strong protective effect of incident asthma.

### 3.4. Significant miRNAs in the Stratified Analysis

We also examined the association separately in both the high vitamin D treatment group and the low vitamin D treatment group (control group). In the high vitamin D treatment group, 32 miRNAs were associated with asthma at age 5 (nominal *p* < 0.05), and of these miRNAs, 22 miRNAs were associated with a higher risk of incident asthma (OR >1) and 10 miRNAs were protective (OR < 1). In particular, there were 7 miRNAs (hsa-miR-3942-5p, hsa-miR-574-5p, hsa-miR-125b-2-3p, hsa-miR-95-3p, hsa-miR-342-3p, hsa-miR-6509-5p, hsa-miR-1294) with OR values greater than 3 and 3 miRNAs (hsa-miR-151a-5p, hsa-miR-6852-5p, hsa-miR-6842-3p) with OR values less than 0.33. Twenty-three significant miRNAs occurring in both the high vitamin D treatment group analysis and the interaction analysis. The results are listed in Table 3. 

Compared with the 32 significant miRNAs in the high vitamin D treatment group, there were only 6 significant miRNAs in the low vitamin D treatment group (control group), of which hsa-miR-505-3p and hsa-miR-340-3p had a higher risk and the others having a protective effect. Four miRNAs, including hsa-miR-7-1-3p, hsa-miR-505-3p, hsa-miR-193b-5p and hsa-miR-5010-5p, overlapped with the significant miRNAs in the interaction analysis. The results are shown in Appendix A.

### 3.5. Validation in Project Viva

We performed a replication analysis in Project Viva on the 23 significant miRNAs identified in both the high vitamin D treatment group and the interaction analysis in VDAART. Because of the small sample size in Project Viva, we focused our validation efforts on both the directionality and effect size. We kept miRNAs with OR ≥ 2 or OR ≤ 0.5 in VDAART and kept miRNAs with OR ≥ 1.5 or OR ≤ 0.67 in Project Viva. Following this methodology, 9 miRNAs were successfully validated in Project Viva and the results are shown in Appendix A. 

Of the 9 validated miRNAs, 6 miRNAs increased the risk of incident asthma and 3 miRNAs had a protective effect on decreasing the risk of incident asthma. For example, hsa-miR-574-5p (Appendix A) was associated with a high risk of incident asthma with a large OR value of 7.2 [95% CI, 1.16 to 44.78] in the high vitamin D treatment group and the OR of the interaction effect is also high (OR = 19.2 [95% CI, 2.38 to 155.11]). These were both validated in Project Viva with OR values of 27.59 [95% CI, 0.19 to 3995.22] and 3.9 [95% CI, 0.15 to 103.06] in the same direction as seen in the VDAART. Hsa-miR-151a-5p (Appendix A) was associated with a protective effect for incident asthma in the high vitamin D treatment group in VDAART with an OR value of 0.15 [95% CI, 0.02 to 0.86] and the effect modification was much stronger with an OR value of 0.05 [95% CI, 0.01 to 0.42]. Both of the values were validated in Project Viva in the same direction with OR values of 0.34 [95% CI, 0.02 to 4.77] and 0.37 [95% CI, 0.01 to 12.54] respectively.

### 3.6. Meta-Analysis of Validated miRNAs in Vitamin D Antenatal Asthma Reduction Trial (VDAART) and Project Viva

To combine the effect in the high vitamin D treatment group of VDAART and high maternal vitamin D group (adequate group) of Project Viva, we performed a meta-analysis on the 9 validated miRNAs. The results are illustrated in Figure 1. All the observed effects of 9 miRNAs were significant in the analysis (nominal *p* < 0.05). Hsa-miR-574-5p had the largest risk for incident asthma with OR = 8.45 [95% CI, 1.52 to 46.97] and hsa-miR-151a-5p had the largest protective effect with OR = 0.19 [95% CI, 0.04 to 0.82]. 

We also performed a meta-analysis on the effect modification and the results were displayed in Appendix A. Both hsa-miR-574-5p and hsa-miR-151a-5p had the largest effects in opposite directions.

### 3.7. MiRNA-Target Gene Network and KEGG Pathway Enrichment Analysis

The miRNA-target gene network was built and displayed in miRNet 2.0. There were 42 KEGG pathways enriched to the 9 validated miRNAs with FDR less than 0.05 and these pathways are shown in Appendix A. Eight signaling pathways were curated and emphasized in Table 4 with known studies about asthma. The top three signaling pathways were Cell Cycle (FDR = 4.8 × 10^−7^), TGF-beta (Transforming Growth Factor beta) Signaling Pathway (FDR = 4.12 × 10^−5^) and ErbB (also known as Epidermal Growth Factor Receptor) Signaling Pathway (FDR = 1.88 × 10^−4^). The target genes involved in these signaling pathways were visualized in Figure 2.

### 3.8. Prediction of Asthma Based on miRNAs

We predicted asthma status based on normalized miRNA counts in the high maternal vitamin D group in Project Viva through a logistic regression model. When using a single miRNA, hsa-miR-574-5p had the best performance with AUROC = 0.83 (Figure 3A). When considering the combination of two miRNAs, hsa-miR-215-5p (AUROC = 0.81) and hsa-miR-370-3p (AUROC = 0.75) performed best with AUROC = 0.86 (Figure 3B).

## 4. Discussion

In our study, we identified several miRNAs that were associated with incident asthma at age 5 years given vitamin D effect modification in VDAART; these miRNAs were replicated in another independent birth cohort, Project Viva. This finding suggests that miRNA may be a pivotal vitamin D related mediator of asthma risk in early childhood. The meta-analysis of the two cohorts strengthened our results and the top miRNA had excellent prognostic power to predict asthma at age 5 years. 

We noted that hsa-miR-574-5p displayed the strongest risk effect in the high vitamin D treatment group of all the validated miRNAs (Figure 1). Sinha and colleagues found hsa-miR-574-5p over expressed by 3.3-fold (*p* = 0.04) in asthmatic patients compared with healthy subjects in the exhaled exosome [43]. Garbacki observed miR-574-5p was down-regulated (fold change = 0.37) with short-term exposure to an allergen but up-regulated (fold change = 13.18) with long-term exposure to allergen in the mouse model of asthma, implying a cell cycle regulatory function [44]. Gomez built a miRNA and mRNA network based on the sputum of patients with asthma, and found the hsa-miR-574-5p module positively correlated with eosinophil counts (*p* = 0.008) and negatively correlated with bronchodilator response (*p* = 0.04) [45]. Moreover, several studies had shown that 25-hydroxyvitamin D (25OHD) or vitamin D3 could alter miRNA expression [46,47]. Hsa-miR-574-5p was down-regulated (fold change = 1.79) in the high vitamin D group in the plasma of pregnant mothers [48]. Together, these studies strongly suggest that hsa-miR-574-5p may be a vitamin D-related mediator and biomarker in asthma. 

Our study also showed that hsa-miR-151a-5p had the strongest protective effect on incident asthma in the high vitamin D treatment group (Figure 1). Francisco-Garcia’s group examined miRNAs in nanovesicles from bronchoalveolar lavage of severe asthmatic patients and they reported that hsa-miR-151a-5p positively correlated with FEV1% (prebronchodilator forced expiratory volume in one second as a percent predicted) (r = 0.48, *p* = 0.03), which confirmed with our analysis [49]. Jorde et al. also observed that hsa-miR-151a-5p was up-regulated in plasma after 1 year of 40,000 IU vitamin D3 per week [46]. In summary, hsa-miR-151a-5p may also be a potential mediator and biomarker. 

We examined the miRNA prognostic ability through a logistic regression prediction model. We noted that with an AUROC = 0.83, hsa-miR-574-5p performed best individually compared to other examined miRNAs (Figure 3A). Hsa-miR-215-5p and hsa-miR-370-3p had the best combinatory prognostic power with AUROC = 0.86 (Figure 3B). Both hsa-miR-215-5p and hsa-miR-370-3p have been shown to affect the inflammatory pathway in lung. Tsuchiya and colleagues found that the expression of hsa-miR-215-5p was elevated (10-fold) in MPA (mucoid *Pseudomonas aeruginosa*) infected CF (cystic fibrosis) lung epithelial cells and implied its pro-inflammatory function [50]. Gupta et al. observed hsa-miR-215-5p differentially expressed (5.46 logFC) in exosome-like vesicles between primary human tracheobronchial cells and a cultured airway epithelial cell line [51]. Hsa-miR-370-3p was involved in inflammatory injury by a long noncoding RNA SNHG16 in LSP (lipopolysaccharides)-induced A549 cells [52]. 

For other validated miRNAs, in our study, many of them were associated with allergic inflammation. MiR-342-3p was associated with allergic airway disease in a mouse population [53] and was also observed to suppress inflammation response in human macrophages THP-1 cells [54]. MiR-122-5p was increased in extracellular vesicles from subjects with asthma [55] and decreased in asthmatic bronchial epithelial cells [56]. MiR-193b-5p played an important role in virus induced lung injury [57] and miR-125b-2-3p might affect the G2/M phase of the cell cycle [58]. 

Using our 9 validated miRNAs, we built a miRNA-target gene network using the data in miRTarBase [29] and performed a KEGG pathway enrichment analysis through miRNet [28]. Eight signaling pathways were identified with an FDR less than 0.05. Of these, cell cycle was the most significant signaling pathway (FDR = 4.8 × 10^−7^) and some studies demonstrated that the airway smooth muscle (ASM) cell proliferation was increased in asthmatic patients [30]. The TGF-beta pathway (FDR = 4.12 × 10^−5^) has been widely investigated and associated with the airway remodeling process in asthma [31,32]. EGFR was observed to be highly expressed in bronchial epithelial cells in asthma [33] and the ErbB signaling pathway (FDR = 1.88 × 10^−4^) plays a key role in mediating airway hyperresponsiveness and remodeling in a chronic allergic mouse model [34]. The Wnt signaling pathway (FDR = 1.23 × 10^−3^) was associated with impaired lung function in childhood asthma [35] and down-regulated by vitamin D, leading to alleviation of airway remodeling [36]. The JAK-STAT (Janus kinase/signal transducers and activators of transcription) signaling pathway (FDR = 2.03 × 10^−4^) was involved in cytokine related regulation and could induce polarization of T helper cells, of which Th2 cells were believed to play an important part in initiating the airway inflammatory response [37,38]. As the central role in the p53 signaling pathway (FDR = 5.77 × 10^−3^), p53 was reported for increased expression in bronchial smooth muscle (BSM) from asthmatic patients and associated with BSM proliferation and mitochondrial biogenesis [39]. T cell receptor signaling pathway (FDR = 3.18 × 10^−2^) contained complex signaling cascades and was crucial for T cell development [40]. The Toll-like receptor signaling pathway (FDR = 4.55 × 10^−2^) has been associated with innate immune system and could induce a pro-inflammatory response, resulting in airway inflammation [41,42]. 

Our study has several limitations most notably the small sample size of both the discovery and validation cohorts. However, through the use of meta-analysis, we demonstrated that all the target miRNAs are significant in their effects. Additionally, most of these miRNAs are associated with allergic inflammation, which is also supported by the KEGG pathway enrichment analysis. Of all the target miRNAs, hsa-miR-574-5p and hsa-miR-151a-5p are the strongest potential mediators and biomarkers of asthma and hsa-miR-574-5p has an excellent prognostic power individually.

## 5. Conclusions

In summary, our findings show that circulating microRNAs are associated with, and predictive of, incident asthma under the effect of vitamin D treatment. Circulating microRNAs may be good biomarkers and mediators of asthma.

## Figures and Tables

**Figure 1 jpm-11-00307-f001:**
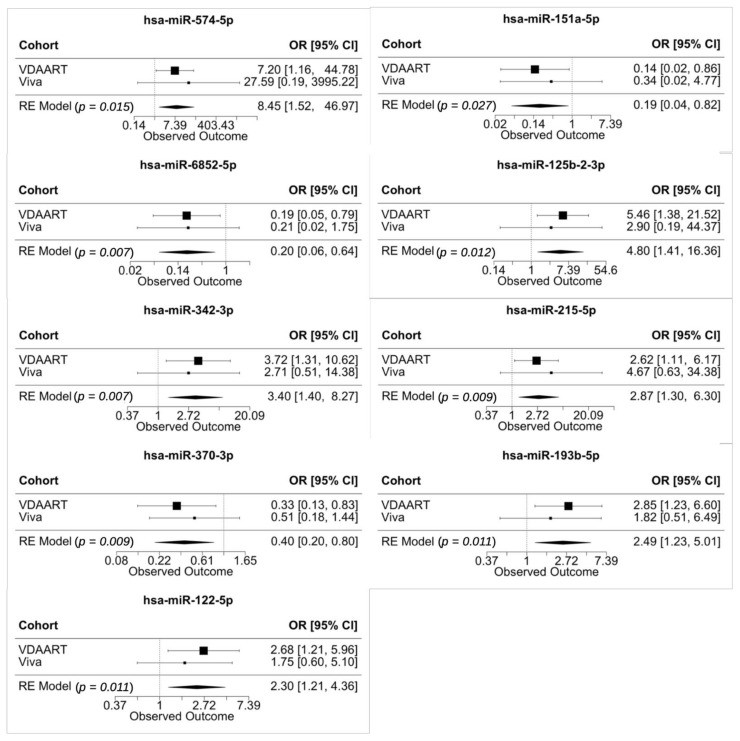
Meta-Analysis of 9 validated miRNAs. MiRNAs are ordered from left top to right bottom in terms of the effect size.

**Figure 2 jpm-11-00307-f002:**
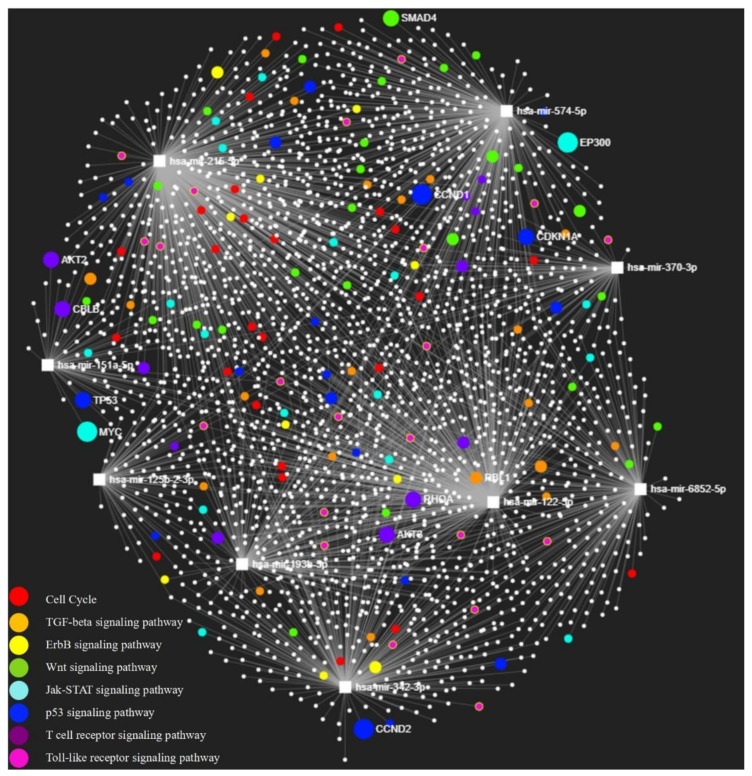
Network of 9 validated miRNAs and target genes. Squares denote miRNAs and circles denote target genes.

**Figure 3 jpm-11-00307-f003:**
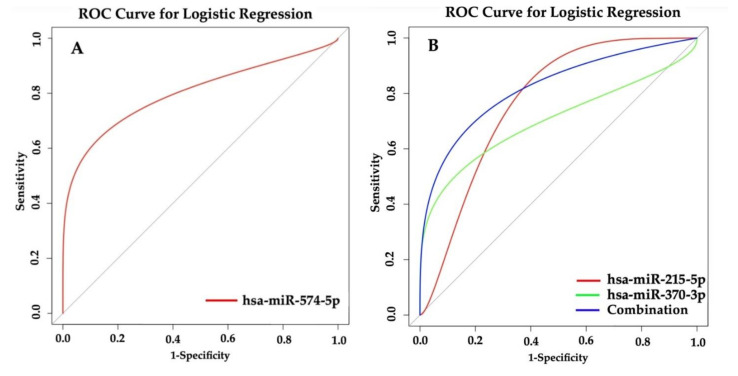
Prediction of hsa-miR-574-5p and the combination of hsa-miR-215-5p and hsa-miR-370-3p in logistic regression model. (**A**) Prediction of hsa-miR-574-5p in logistic regression model. (**B**) Prediction of the combination of has-miR-215-5p and hsa-miR-370-3p in the logistic regression model.

**Table 1 jpm-11-00307-t001:** Baseline characteristics of the Vitamin D Antenatal Asthma Reduction Trial (VDAART) participants at enrollment according to asthma status at age 5 years.

Characteristic	Asthma Status at Age 5 Years		
	Yes (*n* = 31)	No (*n* = 42)	*p* Value
Sex			1.0
Male	21(68%)	28(67%)	
Female	10(32%)	14(33%)	
Race			0.39 *
White	8(26%)	16(38%)	
Black/African American	22(71%)	22(52%)	
Asian	0(0%)	1(2%)	
Other	1(3%)	3(7%)	
Treatment			0.817
4400 IU daily	15(48%)	18(43%)	
400 IU daily	16(52%)	24(57%)	
History of asthma in mother			1.0
Yes	17(55%)	23(55%)	
No	14(45%)	19(45%)	
Maternal age at enrollment	24.51 ± 4.8	25.66 ± 5.8	0.36
Gestation at delivery weeks	37.38 ± 3.57	38.19 ± 2.85	0.3

Data presented as *n* (%) or mean ± standard deviation (SD); IU: international unit; * *p* value from Fisher’s exact test.

**Table 2 jpm-11-00307-t002:** Significant miRNAs by the Vitamin D Treatment Arm Interaction in VDAART.

		95% CI for OR	
miRNA	OR	Lower Bound	Upper Bound
hsa-miR-3942-5p	178.73	4.6	6947.95
hsa-miR-151a-5p	0.05	0.01	0.42
hsa-miR-574-5p	19.2	2.38	155.11
hsa-miR-125b-2-3p	8.62	1.77	41.84
hsa-miR-6852-5p	0.12	0.03	0.58
hsa-miR-7-1-3p	7.81	1.67	36.4
hsa-miR-505-3p	0.13	0.02	0.79
hsa-miR-103a-3p	0.15	0.03	0.81
hsa-miR-1294	5.86	1.55	22.09
hsa-miR-342-3p	5.83	1.63	20.89
hsa-miR-95-3p	5.8	1.93	17.41
hsa-miR-193b-5p	5.55	1.99	15.45
hsa-miR-29a-3p	4.77	1.61	14.11
hsa-miR-331-5p	4.62	1.17	18.18
hsa-miR-146b-3p	4.53	1.23	16.63
hsa-miR-141-3p	4.1	1.36	12.37
hsa-miR-3605-5p	3.87	1.16	12.94
hsa-miR-760	0.3	0.09	0.98
hsa-miR-5010-5p	3.26	1.18	9.05
hsa-miR-122-5p	3.05	1.2	7.74
hsa-miR-215-5p	3.05	1.14	8.13
hsa-miR-370-3p	0.33	0.12	0.94
hsa-miR-29c-3p	2.93	1.25	6.86
hsa-miR-1273h-3p	0.36	0.13	0.96
hsa-miR-144-3p	2.74	1.04	7.23
hsa-miR-1908-5p	0.37	0.14	1
hsa-miR-4732-5p	2.71	1.09	6.75
hsa-miR-339-5p	0.37	0.18	0.76
hsa-miR-483-5p	2.67	1.13	6.28
hsa-miR-214-3p	2.58	1.14	5.85
hsa-miR-671-3p	0.39	0.16	0.95
hsa-miR-342-5p	2.51	1.3	4.85
hsa-miR-150-3p	2.3	1.11	4.77
hsa-miR-134-5p	0.44	0.19	1
hsa-miR-29b-3p	2.26	1.02	4.99
hsa-miR-369-3p	0.45	0.21	1
hsa-miR-130b-5p	0.46	0.23	0.93
hsa-miR-409-3p	0.55	0.31	0.99

**Table 3 jpm-11-00307-t003:** Significant miRNAs in the High Vitamin D Treatment Group in VDAART.

		95% CI for OR	
miRNA	OR	Lower Bound	Upper Bound
hsa-miR-3942-5p	73.05	2.23	2390.05
hsa-miR-574-5p	7.2	1.16	44.78
hsa-miR-151a-5p	0.14	0.02	0.86
hsa-miR-125b-2-3p	5.46	1.38	21.52
hsa-miR-6852-5p	0.19	0.05	0.79
hsa-miR-6842-3p	0.22	0.05	0.86
hsa-miR-95-3p	3.83	1.44	10.14
hsa-miR-342-3p	3.72	1.31	10.62
hsa-miR-6509-5p	3.66	1.02	13.15
hsa-miR-1294	3.47	1.08	11.11
hsa-miR-141-3p	3	1.1	8.18
hsa-miR-370-3p	0.33	0.13	0.83
hsa-miR-331-5p	2.99	1.02	8.76
hsa-miR-424-3p	2.93	1.06	8.1
hsa-miR-193b-5p	2.85	1.23	6.6
hsa-miR-146b-3p	2.82	1.01	7.88
hsa-miR-195-5p	2.71	1.09	6.76
hsa-miR-122-5p	2.68	1.21	5.96
hsa-miR-29a-3p	2.65	1.08	6.54
hsa-miR-127-3p	0.38	0.15	0.97
hsa-miR-215-5p	2.62	1.11	6.17
hsa-miR-30d-3p	0.42	0.18	0.98
hsa-miR-136-3p	0.43	0.19	0.95
hsa-miR-1224-5p	2.26	1.06	4.82
hsa-miR-4732-5p	2.25	1.09	4.66
hsa-miR-642a-3p	2.1	1.01	4.36
hsa-miR-29c-3p	2.05	1.03	4.12
hsa-miR-134-5p	0.5	0.25	1
hsa-miR-150-3p	1.9	1.02	3.53
hsa-miR-342-5p	1.89	1.09	3.29
hsa-miR-130b-5p	0.53	0.3	0.95
hsa-miR-339-5p	0.59	0.35	1

**Table 4 jpm-11-00307-t004:** Significant Kyoto Encyclopedia of Genes and Genomes (KEGG) signaling pathways in association with asthma.

Name	Hits	*p* Value	FDR	Function	Reference
Cell cycle	41	1.44 × 10^−8^	4.8 × 10^−7^	Airway smooth muscle (ASM) proliferation.	[30]
TGF-beta signaling pathway	28	2.47 × 10^−6^	4.12 × 10^−5^	Airway epithelial cells apoptosis, subepithelial fibrosis, airway smooth muscle remodeling, and microvascular changes.	[31,32]
ErbB signaling pathway	27	1.69 × 10^−5^	1.88 × 10^−4^	Airway hyperreactivity and remodeling.	[33,34]
Wnt signaling pathway	36	1.35 × 10^−4^	1.23 × 10^−3^	Airway remodeling.	[35,36]
Jak-STAT signaling pathway	27	2.03 × 10^−4^	1.45 × 10^−3^	Th cell polarization and airway inflammatory response.	[37,38]
p53 signaling pathway	19	1.27 × 10^−3^	5.77 × 10^−3^	Bronchial smooth muscle (BSM) proliferation and mitochondrial biogenesis.	[39]
T cell receptor signaling pathway	22	1.04 × 10^−2^	3.18 × 10^−2^	T cell development and immune system.	[40]
Toll-like receptor signaling pathway	21	1.81 × 10^−2^	4.55 × 10^−2^	Airway inflammation.	[41,42]

## Data Availability

The datasets used and analyzed are available from the corresponding author on reasonable request.

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
