# Peer review of "Circulating MicroRNA: Incident Asthma Prediction and Vitamin D Effect Modification"

_jpm, 2021, doi:10.3390/jpm11040307_

Round 1
Reviewer 1 Report
Asthma is a chronic disease of very large and growing worldwide incidence nowadays and, therefore, good prognostic tools and related preventive approaches are highly valuable. Circulatory miRNAs have been reported in several publications years ago as one of such tools to study asthma, including publications from the group responsible of the here reviewed work. The same applies to indications that Vitamin D intake in pregnant women could substantially reduce asthma in their offspring. The present work reports a study done at this respect with miRNAs isolated and sequenced from blood of a conjoint of children which demonstrate the associations of a series of definite miRNAs, mostly related to allergic inflammation, with asthma and Vitamin D intake, indicating that they could be good biomarkers and have prognostic value for the disease.
The work looks to be carefully made and based, deriving interesting and useful conclusions which, although not absolutely original, are useful and valid for the progression of the field and proposed prognostic approaches. Certainly the used cohorts of participants in the study are not large, but the carefully made meta-analysis seems to indicate that the results and derived immediate conclusions are solid and significant. The identified eight signalling pathways by network analysis and the derived discussion are of particular interest.
Authors should revise the typescript for missing data, as it happens in References 2, 12, 27 and 54, which are incomplete. Also in 264-268 lines, belonging to Discussion, in which explicit reference is made to the work of "Francisco-Garcia's group", a reference that is neither cited there nor in the reference list.
Author Response
Answer to Reviewer 1:
We are sorry for the mistakes of making reference and have revised them all. Some changes are listed as below.
To be completed:
- CDC.gov., National Current Asthma Prevalence (2018). Available online: https://www.cdc.gov/asthma/asthmadata.htm (Accessed on 10 January 2021).
- Zhao, C.; Sun, X.; Li, L., Biogenesis and function of extracellular miRNAs. ExRNA 2019, 1, (1), 1-9.
- Viechtbauer, W., Conducting meta-analyses in R with the metafor package. Journal of statistical software 2010, 36, (3), 1-48.
- Bahmer, T.; Krauss-Etschmann, S.; Buschmann, D.; Behrends, J.; Bartel, S. In miR-122-5p and miR-191-5p are increased in plasma small extracellular vesicles in asthma, ERS International Congress 2019 abstracts, MADRID, Spain, 28 Sep - 2 Oct, 2019.
To be added:
- Francisco-Garcia, A. S.; Garrido-Martin, E. M.; Rupani, H.; Lau, L. C. K.; Martinez-Nunez, R. T.; Howarth, P. H.; Sanchez-Elsner, T., Small RNA Species and microRNA Profiles are Altered in Severe Asthma Nanovesicles from Broncho Alveolar Lavage and Associate with Impaired Lung Function and Inflammation. Noncoding RNA 2019, 5, (4).

Reviewer 2 Report
In this manuscript, Li and colleagues identify several circulating miRNAs that can predict incident childhood asthma in children with recurrent wheeze at the age of 3 years, and this relationship is related to vitamin D treatment. The methods and the results are clearly presented and the conclusion matches the results observed during the study. I recommend this manuscript for publication with minor changes to improve the reader's comprehension:
- Figure 1 is very small and difficult to read. Tables and titles should be bigger.
- Figure 3 is missing the explanation for section A and B in the figure legend.
Author Response
Answer to Reviewer 2:
Thank you for the comments about the Figures and Tables and we fixed all the problems.
For Figure 1, we redrew and rearranged it again to make it much clearer. We also enlarged the tables and titles in the manuscript.
For Figure 3, we redrew it again and it’s much clearer now. For the legend of Figure 3, we added the explanation for section A and B, like “Figure 3. Prediction of hsa-miR-574-5p and the combination of hsa-miR-215-5p and hsa-miR-370-3p in logistic regression model. (A) Prediction of hsa-miR-574-5p in logistic regression model. (B) Prediction of the combination of has-miR-215-5p and hsa-miR-370-3p in logistic regression model.”
We also adjust other tables and figures, please see the revised version of the manuscript.
